## Original Research Article

cell wall; cytoskeleton; micro-indentation; nano-indentation; *Nicotiana tabacum*; statistical modeling.

**Author for correspondence:**
E. Roumeli,
E-mail: eroumeli@uw.edu

# Cell wall and cytoskeletal contributions in single cell biomechanics of *Nicotiana tabacum*

Leah Ginsberg[1], Robin McDonald[1], Qinchen Lin[2], Rodinde Hendrickx[1], Giada Spigolon[3], Guruswami Ravichandran[1], Chiara Daraio[1] and Eleftheria Roumeli[2] 🔘

[1]Division of Engineering and Applied Science, California Institute of Technology, Pasadena, CA 91125, USA; [2]Department of Materials Science and Engineering, University of Washington, Seattle, WA 98195, USA; [3]Biological Imaging Facility, California Institute of Technology, Pasadena, CA 91125, USA

## Abstract

Studies on the mechanics of plant cells usually focus on understanding the effects of turgor pressure and properties of the cell wall (CW). While the functional roles of the underlying cytoskeleton have been studied, the extent to which it contributes to the mechanical properties of cells is not elucidated. Here, we study the contributions of the CW, microtubules (MTs) and actin filaments (AFs), in the mechanical properties of *Nicotiana tabacum* cells. We use a multiscale biomechanical assay comprised of atomic force microscopy and micro-indentation in solutions that (i) remove MTs and AFs and (ii) alter osmotic pressures in the cells. To compare measurements obtained by the two mechanical tests, we develop two generative statistical models to describe the cell's behaviour using one or both datasets. Our results illustrate that MTs and AFs contribute significantly to cell stiffness and dissipated energy, while confirming the dominant role of turgor pressure.

## 1. Introduction

The mechanical properties of plant cells are tightly related to their growth, function, adaptation and survival (Cosgrove, 2016; Milani et al., 2013; Szymanski & Cosgrove, 2009). Enabled by recent developments in mechanical testing and imaging capabilities, the mechanical properties of plant cells have been increasingly studied (Bidhendi & Geitmann, 2019; Burgert, 2006; Geitmann, 2006; Vogler et al., 2015; Wu et al., 2018). The largest body of literature focuses on the properties of the cell wall (CW) and turgor pressure to understand and model the mechanical behaviour of the entire plant cell (Bidhendi & Geitmann, 2019; Braybrook, 2015; Tomos & Leigh, 1999; Weber et al., 2015; Wu et al., 2018). This is in contrast to studies in (wall-less) animal cells, in which the structural roles of the main cytoskeletal filaments, microtubules (MT) and actin filaments (AF), have been established (Gardel et al., 2008; Huber et al., 2015; Janmey, 1991; Pegoraro et al., 2017). In both plant and animal cells, the cytoskeletal filaments form an interconnected network of polymer nanofibres that are responsible for providing structure, and for transducing mechanical signals to assist cell growth, function and development (Durand-Smet et al., 2014; Janmey, 1998). Even though the contributions of cytoskeletal filaments in the mechanical properties of plant cells are of high interest, their direct measurement is challenging due to the presence of the stiff CW in addition to the high turgor pressure inside plant cells. Recent findings show that MTs have a leading role in guiding cellulose deposition in the CW, which indirectly influences the mechanical properties of the CW (Durand-Smet et al., 2014; Paredez et al., 2006). Additionally, rheological tests on plant cells treated to remove their CW show that MTs, in particular, have non-negligible mechanical contributions compared to the CW and turgor pressure (Durand-Smet et al., 2014). Nevertheless, the mechanical contributions of the cytoskeleton in intact plant cells remain unexplored.

During the past decade, advances in mechanical testing instrumentation have enabled remarkable new insights on the importance of plant cell mechanics in plant development. Atomic-force microscopy (AFM), has been used to quantify the elastic modulus of the CW (Braybrook, 2015; Peaucelle et al., 2011), as well as to estimate the turgor pressure (Beauzamy et al., 2015; Vella et al., 2012). AFM results combined with finite element modelling (FEM)

provided evidence that in *Arabidopsis thaliana* (Arabidopsis) pavement cells, the orientation of MTs to mechanical stresses plays a dominant role in guiding cell shape (Sampathkumar et al., 2014). Moreover, AFM has been used to reveal the different elastic properties of the CW in turgid versus plasmolysing solutions in Arabidopsis epidermal cells, highlighting the effects of different stress states in the CW modulus (Braybrook, 2015). Overall, the AFM nano-indentation method allows for the simultaneous acquisition of highly resolved topographical information and mechanical property mapping (Peaucelle et al., 2011; Yilmaz et al., 2020). The applied forces are typically in the pico- to nano-Newton range, and the indenter sizes are a few nanometres wide, which makes the method suitable for highly localised cell properties. When global cell properties are of interest, micron-sized indenters and higher force load cells are required.

Cellular-force microscopy, a method coupling a micro-indentation device with a light microscope, has been applied for such global, cell-level measurements (Nelson, 2011; Routier-Kierzkowska et al., 2012; Vogler et al., 2015; Weber et al., 2015). This apparatus allows for acquisition of micro-indentation data on isolated plant cells, with applied forces in the micro-Newton range. It has been used to obtain direct stiffness measurements of onion epidermal cells which revealed turgor pressure-induced stiffening of the CW and spatial stiffness variations across the tissue surface. In particular, in turgid cells, the surface above the cross-wall junction was softer compared to the middle part of the cells which was stiffer (Routier-Kierzkowska et al., 2012). When used in combination with a computational mechanics model, cellular-force microscopy can be used to extract material properties of subcellular components, such as the elastic modulus of the CW material (Weber et al., 2015).

As more experimental methods to characterise the mechanical properties of plant cells have been adopted, discrepancies arising from comparing results from separate studies have emerged (Bidhendi & Geitmann, 2019; Braybrook, 2015; Vogler et al., 2015). Differences in the sample preparation, loading rate and orientation, indenter shape and size, extent of deformation, models and assumptions for data analysis, on top of variations between samples, justify the literature discrepancies even when the same experimental method is applied (Bidhendi & Geitmann, 2019).

Here, we present a method to compare extracted mechanical properties of plant cells using two techniques: AFM and micro-indentation. This method provides insights into the mechanical contributions of CW, turgor pressure and cytoskeletal filaments in intact *Nicotiana tabacum* Bright Yellow-2 (BY-2) cells, without requiring a complex computational mechanics model of the system. Our multiscale biomechanical assay allows us to probe mechanical properties across multiple length scales which is essential to evaluate the contributions of cytoskeletal fibres, that are a few nanometres in diameter, the CW, which when hydrated is around a micrometre thick, and the bulk protoplasm which is tens of micrometres in diameter and length. To evaluate the effects of turgor pressure, we test cells in solutions of two different osmolarities. In order to isolate the mechanical contribution of the cytoskeleton, we test cells after short exposure to drug treatments that depolymerise MTs and AFs, respectively. We propose a combination of a generative statistical model and a simplified mechanical spring model to analyse the mechanical testing results. This approach allows us to determine the relative stiffness contributions from the CW, MTs, AFs and the rest of the protoplasm, from two independent experimental methods and without the need to create a FEM. To test the stability of our generative statistical model, we perform

an analysis solely based on the micro-indentation data, and then, perform a combined AFM and micro-indentation data analysis. The combined AFM and micro-indentation data analysis more accurately captures the difference in stiffness between the MTs and AFs by taking into account the observed connection between the cytoskeletal filaments and the CW using AFM in conjunction with the micro-indentation data.

## 2. Results and discussion

### 2.1. Cell morphology

We observe the morphology of the unstained BY-2 cells using light microscopy, and upon staining with calcofluor white, we image the cells with confocal laser scanning microscopy (CLSM) (Figure S1a,b). The hydrated CW thickness as visualised in a near-native state from CLSM images is measured to be $0.79 \pm 0.02\,\mu m$ (mean ± standard error), which is similar to values reported for other thin-walled cells in the literature (Moghaddam & Wilman, 1998; Radotić et al., 2012; Yakubov et al., 2016). The observed BY-2 cells are elongated, approximately cylindrical, with cell length and diameter values presented in Figure S1c,d, as measured from light microscopy. The mean observed cell length is $105.43 \pm 3.45\,\mu m$, and the mean observed cell diameter is $39.12 \pm 0.55\,\mu m$, in agreement with prior literature (Čovanová et al., 2013; Sieberer et al., 2009).

BY-2 marker cell lines expressing GFP-tubulin $\alpha$ that visualises MTs (GFP-BY2-$\alpha$), and GFP-AtFim1 to visualise AFs (GFP-BY2-F) were used to study the cytoskeletal changes in response to the selected chemical treatments, which disrupt each of the two networks so that their mechanical property contributions can be isolated. By visualising the cells and their cytoskeleton through CLSM in normal growth media and after short exposures (2 min) to 250 µM latrunculin B (LatB) or 50 µM oryzalin solutions, fluorescent and transmitted light images demonstrate that the short treatment was enough to disrupt the AF and MT networks, without causing plasmolysis or other observable microscopic defects in the cells. Short-term exposures to the drug treatments are chosen to avoid secondary effects of removing components of the cytoskeleton. For example, MTs are known to be linked to the orientation of cellulose microfibrils in the CW, so long-term disruption of MTs could alter the alignment of the cellulose microfibrils, which would in turn inhibit the biological function of the CW (Cosgrove, 2014). Example images of the marker lines before and after short exposures to drug treatments and plasmolysing solution are presented in Figure 1. Additional Z-stacked images of BY-2 marker cells in GM illustrating the transversely oriented (with respect to the cell growth axis) MTs and the more isotropically oriented AFs are provided in Figure S2.

### 2.2. AFM analysis

We subject the wild-type BY-2 cells to AFM tests in GM (growth media) and PS (plasmolysing solution) to evaluate indentation moduli of the CW in solutions of different osmotic pressures. To determine any effects of MT and AF removal on the elastic properties of the CW, we subject the cells to short treatments of oryzalin (Durand-Smet et al., 2014), or LatB (Durst et al., 2014; Maisch et al., 2009), which are added to the GM or PS (see Section 2). There are six testing conditions: GM, GM–MT, GM–AF, PS, PS–MT, PS–AF, where -MT indicates the oryzalin treatment which depolymerises MTs, and -AF indicates the LatB treatment which removes AFs.

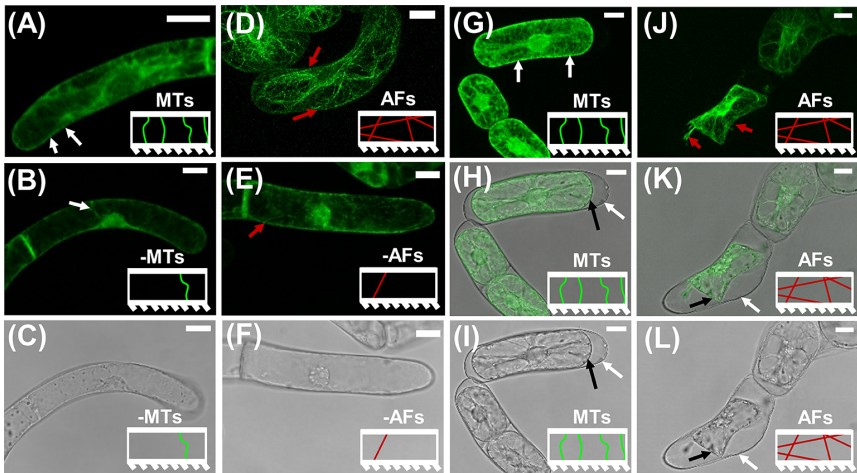

**Fig. 1.** CLSM images on BY-2 marker lines to visualise the effects of short duration drug treatments on MTs and AFs. The model representation of MTs and AFs are included as insets in each image panel. (a)–(c) CLSM images of GFP-BY2-$\alpha$ cells in growth media-based solutions. White arrows point to larger bundles of MTs that are visible near the CW. (a) Fluorescence image in pure growth media. (b) Fluorescence image after exposure to growth media-based oryzalin solution. (c) Corresponding transmission light image for (b), showing no visual morphological change in the cell as a result of the short-term exposure to the drug treatment. (d)–(f) CLSM images of GFP-BY2-F cells in growth media-based solutions. Red arrows point to visible larger bundles of polymerised AFs. (d) Fluorescence image in pure growth media. (e) Fluorescence image after exposure to growth media-based LatB solution. (f) Corresponding transmission light image for (e), showing no evident morphological change in the cell as a result of the treatment. (g)–(i) CLSM images of GFP-BY2-$\alpha$ cells in sorbitol. (g) Fluorescence image. (h) Combined fluorescence and transmission light image of (g and i). (i) Transmission light image. (j)–(l) CLSM images of GFP-BY2-F cells in sorbitol. (j) Fluorescence image. Red arrows point to visible larger bundles of polymerised AFs. (k) Combined fluorescence and transmission light image of (j and l). (l) Transmission light image. In panels (h, i, k and l) white arrows point to CWs and black arrows point to plasma membranes, which have retracted from the CW. All scale bars are 20 $\mu$m.

Considering that cellulose fibrils, which comprise the structural backbone of the CW, are known to be organised in 5–50 nm-thick bundles (Moon et al., 2011), and are immersed in a continuous, heterogeneous matrix of hemicellulose, pectin and proteins, we use an AFM tip with a spherical bead of 1 $\mu$m diameter to probe the bulk behaviour of the CW (Braybrook, 2015). The average indentation depth for turgid cells is 84.7 ± 4.7 nm, which is shallow enough (with respect to the hydrated total CW thickness) to assume that the observed mechanical response is solely from the CW (Braybrook, 2015; Milani et al., 2013; Radotić et al., 2012; Sampathkumar et al., 2014). Plasmolysis of cells removes turgor pressure and enables deeper nano-indentations of the CW, without probing the protoplasm. The average indentation depth of plasmolysed cells is 217.0 ± 45.8 nm, which is approximately 20% of the hydrated CW thickness. This indentation depth is shallower, with respect to cell size, than other literature-reported indentations aiming to isolate the response of the CW alone (Peaucelle et al., 2011). In all cases, the extracted CW modulus is reflective of the mechanical properties of the top layers of the CW material since that is the area of the CW which we are stressing with a shallow indentation force. Typical force-indentation and retraction data with an overlaid Hertz model fit is presented in Figure 2a, with example images from cells in each solution in the insets. The Young's modulus results, separated by treatment, are presented in Figure 2b,c.

Our data show that cells in all growth media-based solutions have a CW Young's modulus ranging from 0.65 to 15.2 MPa, while in all plasmolysing solutions cells have a modulus ranging between 0.03 and 0.49 MPa. The non-parametric Kolmogorov–Smirnov test reveals that there is a significant difference between the CW moduli of cells tested in GM versus those in PS, with $p = 4.4 \times 10^{-16}$. Furthermore, the removal of AFs results in the largest reduction of Young's modulus, in cells in both GM and PS. This observation suggests that there must be a connection between AFs and the CW that is detectable from the conducted AFM tests which probe local, exterior layers of the CW responses. The depolymerisation

of MTs also reduces the Young's modulus in GM, but does not make a significant difference in the cells tested in PS. Specifically, in absence of drug treatments in the GM solution, we observe a CW modulus of $E_{GM} = 6.3 \pm 1.1$ MPa, which is significantly different from the moduli in GM–MT ($E_{GM-MT} = 4.2 \pm 0.6$ MPa) and GM–AF ($E_{GM-AF} = 2.2 \pm 0.1$ MPa) treatments, with $p$-values of 0.049 and $5.8 \times 10^{-5}$, respectively. The GM–MT and GM–AF treatments also lead to significantly different CW moduli, with a $p$-value of $1.1 \times 10^{-4}$. The CW modulus in pure PS treatment is $E_{PS} = 270 \pm 60$ kPa and is significantly different from the PS–AF ($E_{PS-AF} = 130 \pm 30$ kPa) treatment, with a $p$-value of 0.0015. The CW moduli in PS–MT ($E_{PS-MT} = 300 \pm 30$ kPa) and PS–AF conditions are also significantly different, with a $p$-value of $1.1 \times 10^{-4}$. Thus, from the nano-scale measurements, we draw two main conclusions: (a) the biggest changes in the CW modulus are caused by the changes in turgor pressure, and we confirm experimentally that the higher internal pressure stiffens the CW through stress, as predicted in (Cosgrove, 2016) and (b) there is an evident interconnection between the cytoskeleton and CW, which is manifested through CW softening in response to the cell being subjected to drug treatments targeting the cytoskeleton.

### 2.3. Micro-indentation experiments and generative spring model

For the micro-indentation experiments, BY-2 cells are tested in the same testing conditions as in the AFM experiments (see Section 2). Representative force curves and images from the micro-indentation test are provided in Figure 3. The imaging capabilities during mechanical testing allow us to clearly observe plasmolysis effects (Figure 3c) where a cell in PS has the plasma membrane peeled away from the outer CW and the protoplasm retracted.

We calculate the initial effective stiffness by a linear fit to the first 1 $\mu$m of indentation data after contact is initiated. This indentation depth is close to the thickness of the CW as measured from CLSM. Hence, the recorded mechanical response of the cell can

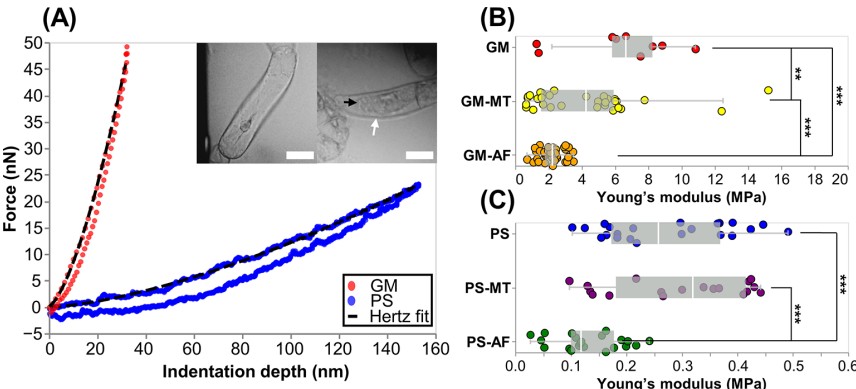

**Fig. 2.** (a) Typical AFM force-indentation and retraction data from a cell in GM and in PS with Hertz fit to indentation data overlaid. Insets show corresponding images of cells in the AFM test in GM (left) and PS (right). Arrows point to CW (white) and retracted plasma membrane (black). Scale bars are 40 μm. (b) Plot of indentation moduli for cells in all drug treatments in GM. (c) Plot of indentation moduli for cells in all drug treatments in PS. Note the difference in scales between (b) and (c). Each point in the plot represents an indentation test. In each test condition, there are $n \geq 9$ tests from five biological replicates. Stars indicate significant differences in distributions according to the nonparametric Kolmogorov–Smirnov test. $** p < 0.05$, $*** p < 0.01$.

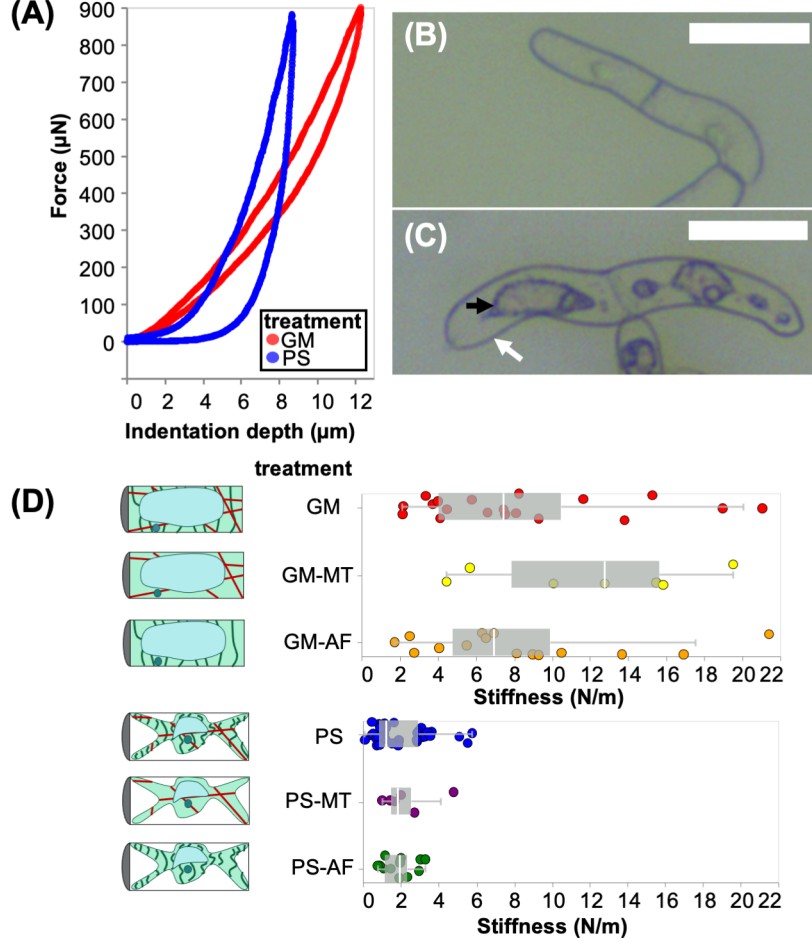

**Fig. 3.** (a) Representative force-indentation and retraction data obtained in micro-indentation experiments on cells in GM (growth media) and in PS (plasmolysing solution). (b) Image of BY-2 cells in GM taken from optical microscope of the micro-indentation testing apparatus. (c) Image of BY-2 cells in PS taken from optical microscope in micro-indentation testing apparatus with arrows pointing to the CW (white) and retracted plasma membrane (black). Scale bars are 100 μm. (d) Box and whiskers plot overlayed on initial cell stiffness data in each test condition. Each point in the plot represents an indentation test on a different cell ($n \geq 6$).

be attributed to a combination of the CW and the underlying protoplasmic materials. In the plots of Figure 3d, the quantiles of each dataset are overlaid as a boxplot on the stiffness data. In Figure S3a, the empirical cumulative distribution functions (ECDFs) are

shown, which enable the visualisation of the distribution of cell stiffness measurements across treatments.

The evident increase in stiffness observed in cells tested in a solution of lower osmotic pressure illustrates the dominant effects

of turgor pressure, in comparison to all other effects under consideration. Specifically, we note two distinct groupings in the measured distributions: cells in GM suspensions ($k_{GM,all}$ = 8.95 ± 0.86 N/m), and cells in PS suspensions ($k_{PS,all}$ = 1.99 ± 0.18 N/m). The *p*-value which separates the stiffness of cells in GM and PS conditions is $p = 7.22 \times 10^{-17}$. BY-2 cell stiffness measurements previously reported in literature are in good agreement with the average cell stiffnesses shown in Figure 3d (Felekis et al., 2012; Weber et al., 2015). Specifically, stiffness ranges of 10–33 N/m were reported from (Felekis et al., 2012) for turgid cells, while back-calculated values of 10 and 5 N/m can be extracted from (Weber et al., 2015) for BY-2 cells in water and 0.2 M mannitol, respectively. The trend of turgor pressure increasing the stiffness of the cell is also reflected in measurements by (Weber et al., 2015). There was no statistically significant difference between any of the groups within the GM or PS categories.

The stiffness results support the dominance of turgor pressure on the cell stiffness, which is in agreement with our aforementioned AFM analysis, and literature (Routier-Kierzkowska et al., 2012). Directly from the experimental results, we conclude that an increase in turgor pressure results in a dramatic increase in cell stiffness at multiple measurement scales. Beyond this conclusion, we aim to extract insights for the mechanical properties of other subcellular structures, especially the cytoskeletal filaments. Although there was no statistically significant difference between the measured stiffnesses of the cells in all the different treatments, we propose using a mechanical model to elucidate trends and effects caused by the different treatments on the mechanical contributions of sub-cellular components.

We apply a generalised two-spring model, which was introduced in our prior work (Roumeli et al., 2020), to separate the stiffness contributions from the CW and the protoplasm. In the two-spring model (Figure S4), the mechanical response of a cell to micro-indentation experiments is modeled as two springs acting in series. Previous literature reports modeled the mechanical response of a cell as a single spring by reporting apparent cell stiffness (Beauzamy et al., 2015). In our model, the apparent cell stiffness is separated into contributions from the CW and protoplasm, using spring constants $k_{CW}$ and $k_{prot}$, respectively.

$$k_{total} = \frac{k_{CW} k_{prot}}{k_{CW} + k_{prot}}. \tag{1}$$

This simplified model of the cell response relies upon assumptions about the structure and materials that constitute the cell. The CW and protoplasmic materials are assumed to behave as homogeneous, isotropic, linear elastic materials at shallow indentation depths, and any nonlinear behaviours, such as viscosity, adhesion, or plasticity are not captured by the model. The stress in the cell away from the indenter is assumed to be negligible for shallow indentations (Boussinesq, 1885). The interpretation of stiffness with respect to subcellular structures is somewhat controversial due to the heterogeneity, directionality and variability inherent to biological systems.

### 2.4. Analysis of micro-indentation data

As a first iteration on the micro-indentation results, we assume that the CW stiffness remains constant across drug treatments, but not across osmotic solutions. This assumption is in accordance with the observation from our AFM data that changes in the CW elastic modulus caused by depolymerising MTs and removing AFs

were much less significant than the change caused by different osmotic pressures. In the succeeding analysis section, which combines results from the AFM and micro-indentation experiments, we will remove this assumption, and analyse results from both experiments simultaneously.

An illustration of the spring model adapted to each test condition is presented in Figure S5. To extract the stiffness contributions from the MTs and AFs, we model them as springs in parallel to the rest of the protoplasmic response, with coefficients $k_{MT}$ and $k_{AF}$. To account for the change in the protoplasm in different osmotic conditions, the protoplasmic response is differentiated between cells in GM and cells in PS. The GM is a hypotonic solution that allows the cell to maintain turgor pressure, nutrients to flow into the cell, and the cell to expand. The PS is a hypertonic solution since the osmotic pressure of a solution that causes plasmolysis (instant response visible through microscopic views of both mechanical testing methods, see Figure 1g–l, inset in Figures 2a and 3c) must be higher than the osmotic pressure of the cell. The spring constants $k_{hypo}$ and $k_{hyper}$ represent the stiffness contribution from all protoplasmic components in GM and PS, respectively, excluding the MTs and AFs, which are already represented by $k_{AF}$ and $k_{MT}$ in the spring model.

In total, we have six spring stiffnesses that are calculated though our analysis: $k_{CW,hypo}$, $k_{CW,hyper}$, $k_{hypo}$, $k_{hyper}$, $k_{MT}$ and $k_{AF}$. We also have six measurements of the effective stiffnesses from the six testing conditions: GM, GM–MT, GM–AF, PS, PS–MT and PS–AF. Since we have an equal number of variables and datasets, a unique solution to the system of effective stiffness equations is possible. However, the equations are nonlinear and cannot be solved analytically. To tackle this, we develop a generative statistical model (Betancourt, 2019; Bois, 2018).

Generative statistical models are used to build a posterior probability distribution $g(\theta|y)$, which is the probability that a set of parameters $\theta$ describes the given experimental data $y$. Here, we are interested in the posterior probability distribution for the parameters $\theta = \{k_{CW,hypo}, k_{CW,hyper}, k_{hypo}, k_{hyper}, k_{MT}, k_{AF}\}$ given the dataset $y = \{k_{GM}, k_{GM-MT}, k_{GM-AF}, k_{PS}, k_{PS-MT}, k_{PS-AF}\}$, where each variable in $y$ represents a set of measurements of the stiffness from the selected treatment. Thus, the posterior probability distribution details the probability that a set of deconvoluted subcellular stiffness constants describe the observed experiments. We use six separate posterior probability distributions, all of which are dependent on each other through the subcellular stiffness constants $\theta$. Using Bayes' theorem, we solve for $g(\theta|y)$, using the likelihood of observing our experimental data given a selected set of parameters, $f(y|\theta)$, and prior information about our parameters of interest, $g(\theta)$. The likelihood is defined separately for each treatment using a Gaussian distribution, and the prior distribution is defined empirically (see Supplementary Materials).

We model the overall stiffness of each cell measured in each test condition using the two-spring model in Figure S4. Adaptations of equation (1) for each testing condition gives the relationship between the mean overall stiffness in each treatment (μ) and the stiffness of each sub-cellular component ($k$). The equivalent equations for the spring stiffness are:

$$\mu_{GM} = \frac{k_{CW,hypo}(k_{hypo} + k_{AF} + k_{MT})}{k_{CW,hypo} + k_{hypo} + k_{AF} + k_{MT}}, \tag{2}$$

$$\mu_{GM-MT} = \frac{R_{GM-MT} k_{CW,hypo}(k_{hypo} + k_{AF})}{R_{GM-MT} k_{CW,hypo} + k_{hypo} + k_{AF}}, \tag{3}$$

$$\mu_{\text{GM-AF}} = \frac{R_{\text{GM-AF}}k_{\text{CW,hypo}}(k_{\text{hypo}} + k_{\text{MT}})}{R_{\text{GM-AF}}k_{\text{CW,hypo}} + k_{\text{hypo}} + k_{\text{MT}}}, \tag{4}$$

$$\mu_{\text{PS}} = \frac{k_{\text{CW,hyper}}(k_{\text{hyper}} + k_{\text{AF}} + k_{\text{MT}})}{k_{\text{CW,hyper}} + k_{\text{hyper}} + k_{\text{AF}} + k_{\text{MT}}}, \tag{5}$$

$$\mu_{\text{PS-MT}} = \frac{R_{\text{PS-MT}}k_{\text{CW,hyper}}(k_{\text{hyper}} + k_{\text{AF}})}{R_{\text{PS-MT}}k_{\text{CW,hyper}} + k_{\text{hyper}} + k_{\text{AF}}}, \tag{6}$$

$$\mu_{\text{PS-AF}} = \frac{R_{\text{PS-AF}}k_{\text{CW,hyper}}(k_{\text{hyper}} + k_{\text{MT}})}{R_{\text{PS-AF}}k_{\text{CW,hyper}} + k_{\text{hyper}} + k_{\text{MT}}}, \tag{7}$$

where the ratios $R$ are all equal to 1 for this initial analysis, since we assume that the removal of MTs or AFs has no effect on the stiffness of the CW.

With these six equations, we transform the means of each of the six treatments to identify the six parameters of interest, $\theta = \{k_{\text{CW,hypo}}, k_{\text{CW,hyper}}, k_{\text{hypo}}, k_{\text{hyper}}, k_{\text{MT}}, k_{\text{AF}}\}$. To optimise the posterior distributions for all six parameters of interest simultaneously, we combine the six separate posterior distributions into one objective function and add six coefficients ($a$, $b$, $c$, $d$, $e$ and $f$) that will be optimised concurrently. These six coefficients are used to balance the final objective function, in absence of finding the true solution to the system of six equations. Mathematically, we maximise $F$ over $a$, $b$, $c$, $d$, $e$, $f$, and $\theta$:

$$F = a * g_{\text{GM}}(\theta|y) + b * g_{\text{GM-MT}}(\theta|y) + c * g_{\text{GM-AF}}(\theta|y) + d * g_{\text{PS}}(\theta|y)$$
$$+ e * g_{\text{PS-MT}}(\theta|y) + f * g_{\text{PS-AF}}(\theta|y). \tag{8}$$

Projections of the objective function into two-dimensional space are presented in Figure S6, illustrating the correlations between each pair of the six stiffness parameters.

Coefficients $a$, $b$, $c$, $d$, $e$ and $f$ are weights multiplied in front of the posterior distributions for each of the six parameters of interest. The weights should all sum up to unity. We also add constraints on the size of the coefficients and the size of the spring constants to ensure that all components are included, and none dominate the optimisation function

$$a + b + c + d + e + f = 1, \tag{9}$$

$$0.05 \leq a, b, c, d, e, f \leq 0.5, \tag{10}$$

$$0.01 \leq \theta \leq 100. \tag{11}$$

Using a trust-region constrained optimisation method, we can find the parameters $\theta$ that maximise the posterior distributions in all treatments, under the above constraints. We can also construct a credible region by calculating the Hessian at the optimised point in parameter space. The optimum parameter values (also known as the maximum a posteriori or MAP parameter values) and credible region (which contains approximately 68% of the total probability) are reported in the following discussion as $k_{\text{MAP}} \pm \sigma_{\text{MAP}}$ (mean ± standard deviation), and the results are visualised in Figure S6 as a red 'x' overlaying the projections of the combined posterior distributions.

By optimising the objective function from the combined posterior distributions, we can decouple the relative stiffness contributions from the six identified subcellular components of interest, and the results are in line with previous reports and predictions in literature (Cosgrove, 2016; Durand-Smet et al., 2014; Routier-Kierzkowska et al., 2012). The contribution from the protoplasm without MTs and AFs in hypotonic conditions is the highest component evaluated ($k_{\text{hypo}} = 42.03 \pm 2.01$ N/m), and about four times

greater than that of the protoplasm without MTs and AFs in hypertonic conditions ($k_{\text{hyper}} = 9.68 \pm 1.50$ N/m). This is in agreement with literature which shows that turgor pressure supplies most of the stiffness for turgid cells in compression (Routier-Kierzkowska et al., 2012).

High turgor pressure in hypotonic conditions stresses the CW, making its response to compression appear stiffer. The stiffness of the CW from AFM indentations in GM was 5.5 times greater than the stiffness of the CW in PS. Without the inclusion of these results in the current analysis, the model predicts that the CW stiffness in hypotonic conditions ($k_{\text{CW,hypo}} = 12.43 \pm 0.68$ N/m) is about twice as high as the CW stiffness in hypertonic conditions ($k_{\text{CW,hyper}} = 6.95 \pm 0.32$ N/m). The results from this analysis represent both material and structure of the CW, while the AFM results probe only the CW material. The fact that both quantities are higher in GM than in PS could be merely a result of CW stiffening under high turgor pressure, or it could be a result of both CW strain-stiffening and an increase in the bending rigidity of the CW under a higher turgor pressure (or another unknown geometric or structural change in the cell under pressure).

The credible regions for the relative stiffness contribution from AFs ($k_{\text{AF}} = 11.81 \pm 4.69$ N/m) and MTs ($k_{\text{MT}} = 6.82 \pm 2.48$ N/m) overlap, and are on the same order of magnitude as the CW stiffness. This result clearly demonstrates that the cytoskeleton is an important structural component for the cell.

### 2.5. Combined analysis of AFM and micro-indentation data

Literature results confirm that MTs and AFs are physically connected to the CW, and thus the removal of these filaments should affect the mechanical behaviour of the CW (Szymanski & Cosgrove, 2009). Our AFM experiments support this fact, as we measure that the CW stiffness is indeed affected by the removal of cytoskeletal filaments with drug treatments, albeit appreciably less than the effect of altering the osmotic pressure of the solution. In this part of our analysis, we introduce the observed effect of the drug treatments on the stiffness of the CW in our generative statistical model using ratios of the mean measured CW stiffnesses from the AFM tests. We calculate the CW stiffness values from the AFM data using a linear interpolation of the first 10% of the maximum force data after contact is detected. The use of ratios instead of absolute values of stiffness is selected to overcome discrepancies in measuring the same properties with different experimental techniques, which have also been reported in literature (Bidhendi & Geitmann, 2019; Wu et al., 2018).

The ratios used are:

$$R_{\text{GM-MT}} = \frac{k_{\text{AFM, GM-MT}}}{k_{\text{AFM, GM}}}, \tag{12}$$

$$R_{\text{GM-AF}} = \frac{k_{\text{AFM, GM-AF}}}{k_{\text{AFM, GM}}}, \tag{13}$$

$$R_{\text{PS-MT}} = \frac{k_{\text{AFM, PS-MT}}}{k_{\text{AFM, PS}}}, \tag{14}$$

$$R_{\text{PS-AF}} = \frac{k_{\text{AFM, PS-AF}}}{k_{\text{AFM, PS}}}, \tag{15}$$

where $k_{\text{AFM, treatment}}$ is the mean indentation stiffness measured in the specified treatment in AFM experiments. The use of these ratios allows us to introduce the change in CW stiffness, as observed in the AFM experiments, in the micro-indentation analysis,

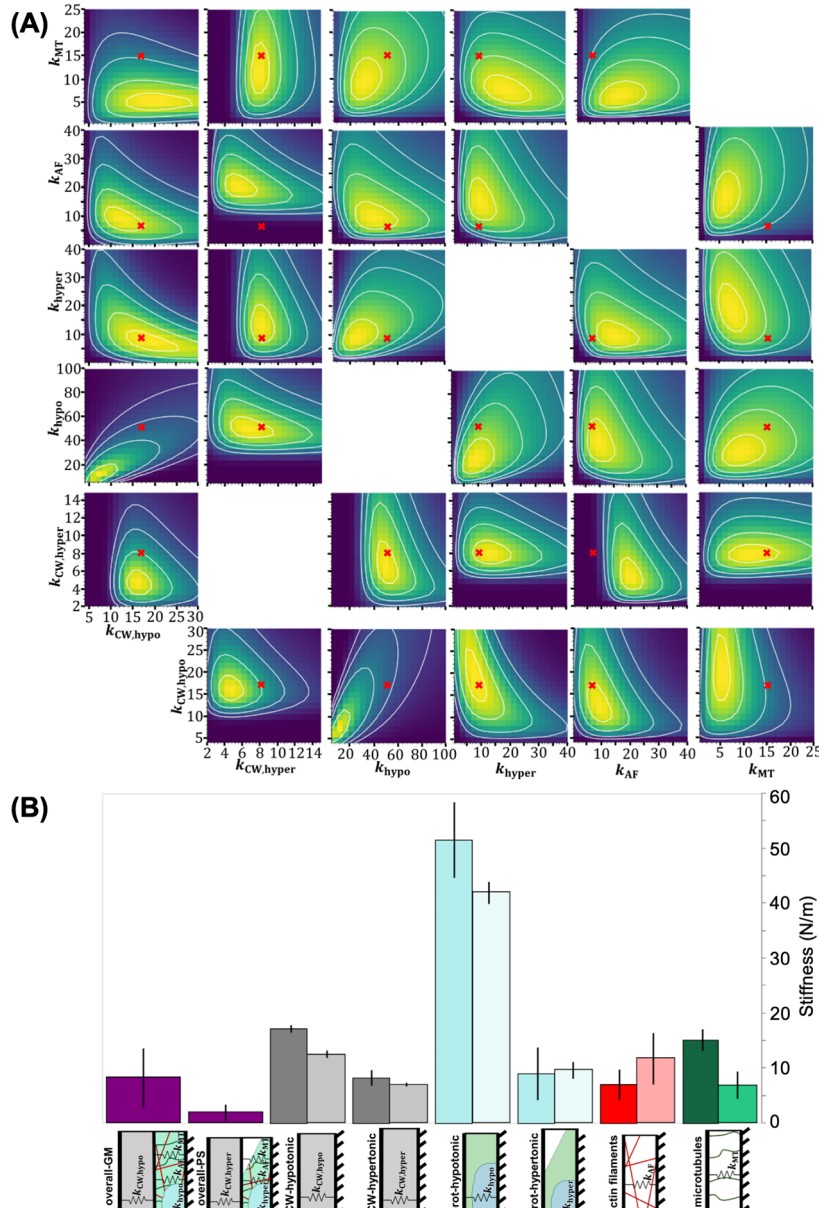

**Fig. 4.** (a) Contours of six-dimensional posterior distribution projected in two-dimensional space using modified stiffness equations to account for the change in CW stiffness from drug treatments, as observed in the AFM tests. Red 'x' marks the point which maximises the posterior distribution, known as the maximum a posteriori (MAP) estimate. The white lines represent 1, 2, 3 and 4 standard deviations from the center of the distribution. All stiffness units are N/m. (b) Comparison of stiffness values for the overall stiffnesses and each deconvoluted subcellular component in both analyses. From left to right, each bar represents: overall cell stiffness measured in hypotonic and hypertonic solutions with no added drug treatments (purple), deconvoluted CW stiffness in hypotonic and hypertonic solution (grey), deconvoluted stiffness from the cytosol, vacuole, and other organelles in hypotonic and hypertonic solution (blue), deconvoluted stiffness from actin filaments (AFs) (red) and deconvoluted stiffness from the microtubules (MTs) (green). The left and darker colored bars represent results from the analysis with AFM and micro-indentation results combined. The right and lighter colored bars represent results from the original analysis that only considered the micro-indentation data. Error bars represent standard deviation, so that the range covered by the error bars represents 68% of the total probability distribution for each stiffness.

without adding additional parameters to optimise. Since the testing conditions are the same in both experiments, we assume that the ratios of mean CW stiffnesses are the same in both sets of experiments. The ratios presented in equations 12–15 are substituted into equations 4–7 to produce the equivalent equations for spring stiffness which include the effect of removing cytoskeletal filaments on the CW stiffness. With these six equations, we can transform the means of each of the six treatments to identify the same six parameters of interest as in the initial analysis of the micro-indentation data, $\theta = \{k_{CW,hypo}, k_{CW,hyper}, k_{hypo}, k_{hyper}, k_{MT}, k_{AF}\}$.

Again, we maximise the objective function from equation (8), $F$, over $\theta$ and the weighting coefficients $a$, $b$, $c$, $d$, $e$ and $f$. Projections of the objective function into two-dimensional space using the modified spring stiffness equations is presented in Figure 4a. Using the same trust-region constrained optimisation method, we find the parameters that maximise the combined posterior distributions in all treatments, using the modified stiffness equations. These points are marked by a red 'x' in Figure 4a. The MAP parameter values and credible regions from both analysis methods are visualised in Figure 4b for comparison.

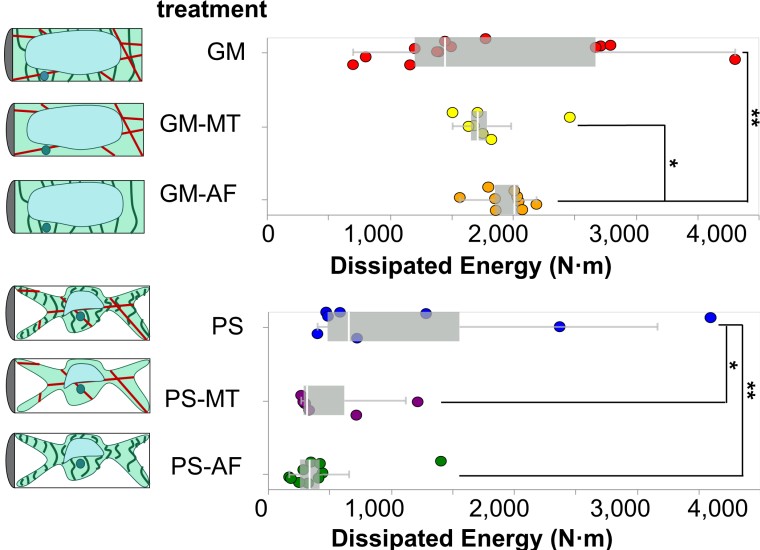

**Fig. 5.** Box and whiskers plot overlaid on energy dissipation data from cells in each test condition. Each point in the plot represents an indentation test on a different cell ($n \leq 6$). Stars indicate significant differences in distributions according to the nonparametric Kolmogorov–Smirnov test. $* \; p < 0.1$, $** \; p < 0.05$.

The results presented in Figure 4b confirm that the same trends hold true for results from both analyses. Firstly, in the combined analysis of the AFM and micro-compression experiments, the CW in hypotonic conditions ($k_{CW, hypo} = 17.22 \pm 0.68$ N/m) is about twice as stiff as the CW in hypertonic conditions ($k_{CW, hyper} = 8.27 \pm 1.40$ N/m), confirming the effective CW stiffening upon exertion of turgor pressure. This is 40 and 20% higher compared to our prior analysis, respectively for turgid and plasmolysed conditions. Secondly, the contribution from the protoplasm without MTs and AFs in hypotonic conditions is the highest component evaluated ($k_{hypo} = 51.55 \pm 6.86$ N/m), and it is about five times greater than in hypertonic conditions ($k_{hyper} = 9.02 \pm 4.78$ N/m). The protoplasmic stiffness without MTs and AFs in the hypotonic treatments is approximately 20% higher compared to our prior analysis, and in the hypertonic treatment the values are roughly the same in both analyses (approximately 7% difference). The MT stiffness ($k_{MT} = 15.16 \pm 1.96$ N/m) is nearly twice as high as the AF stiffness ($k_{AF} = 7.06 \pm 2.72$ N/m), which is the opposite of the trend observed in the previous analysis. The trend revealed in the present analysis is in agreement with literature, which reports that the rheological properties of *Arabidopsis* protoplasts with depolymerised MTs are significantly lower than those of untreated protoplasts, while AF removal does not significantly change the protoplast rheological properties (Durand-Smet et al., 2014). In our combined analysis, the effect of the cytoskeletal filaments on stiffness is more accurately captured, since the effect of the drug treatments on the CW is included. We note that our experiments are conducted in walled cells, while literature-reported results were obtained from wall-less protoplasts (Durand-Smet et al., 2014), which constitutes a significant difference amongst the two approaches. Nevertheless, we observe the same trend of higher stiffness contribution from MTs as compared to AFs, and, more importantly, that both those contributions are comparable to the CW stiffness.

### 2.6. Analysis of dissipated energy

In order to evaluate the contributions of the subcellular elements on the dissipated energy during the micro-indentation tests, we calculate the area between the indentation and retraction curves

in experiments reaching a force threshold of 800 μN. This area represents the energy dissipated by the cell during the indentation experiment. The dissipated energies for cells in each treatment are presented with overlaid boxplots in Figure 5. To visualise the distribution of measurements, see the ECDFs of the dissipated energy data for cells in each treatment in Figure S3b. Similarly to our stiffness analyses, we find that the osmotic treatment results in the most dominant differences across all studied conditions. The two groups of cells in GM ($W_{GM,all} = 1840 \pm 119$ N/m²) and PS ($W_{PS,all} = 736 \pm 177$ N/m²) are significantly different with a $p$-value of $1.46 \times 10^{-7}$. Therefore, we conclude that turgor pressure affects not only the ability of the cell to store elastic energy, but also to dissipate energy.

The results show that in GM conditions the removal of MTs does not produce a detectable difference in the dissipated energy upon indentation. With turgor pressure removed, in PS conditions, the effects of MTs in providing dissipating energy modes to the cell are revealed. Upon removal of MTs, the average dissipated energy is reduced by a significant amount in the PS treatments, from is $W_{PS} = 1240 \pm 438$ N/m² in the pure PS treatment, to $W_{PS-MT} = 523 \pm 142$ N/m² in PS–MT treatment, with a corresponding $p$-value of 0.091. Literature reports show that MTs contribute to energy dissipation by buckling (Li, 2008; Soheilypour et al., 2015).

The removal of AFs produces significantly different dissipated energies in both osmotic conditions. In particular, it leads to the highest average dissipated energy in GM conditions ($W_{GM-AF} = 1940 \pm 59$ N/m²), and the lowest at PS conditions ($W_{PS-AF} = 458 \pm 132$ N/m²). Those values are significantly separated from the dissipated energies of cells in pure GM and PS conditions by $p$-values of 0.012 and 0.024, respectively. Given that our results show that MTs contribute energy dissipation modes to the cell, we postulate that the removal of AFs causes an increase in the average dissipated energy in turgid cells by allowing more unrestricted movement and buckling of the MTs, since the two networks are interpenetrated in the protoplasm. This is in agreement with literature which shows that AFs act as a soft mesh that restricts MTs buckling under load in *in vitro* systems (Ricketts et al., 2018). Finally, given that our results show that the removal of either AFs or MTs in plasmolysed cells causes a significant reduction of the dissipated energy in

comparison to cells with intact MTs and AFs, we conclude that both MTs and AFs must contribute to the energy dissipation mechanisms of the cell.

## 3. Conclusions

In this work, we characterise the mechanical properties of *Nicotiana tabacum* cells, aiming to evaluate the distinct contributions from the CW, the protoplasm and the two main cytoskeletal components, MTs and AFs. We apply a multiscale biomechanical assay comprised of AFM and micro-indentation experiments, and test the cells in different osmotic solutions to control osmotic pressure, and in drug treatments that selectively remove either MTs or AFs. We then propose a generative statistical model that utilises stiffness measurements from the two independent experimental methods to deconvolute the relative contributions of the subcellular components to the cell stiffness. Using two sets of assumptions in our statistical model, our main conclusions are consistent, validating the model and the extracted trends.

Our results confirm that the cytoskeleton contributes significantly to the stiffness and dissipated energy of *Nicotiana tabacum* cells in indentation. Using the model which takes into account the contribution of MTs and AFs to CW stiffness, we find that the MT network contributes nearly double the amount of stiffness as the AF network, in agreement with studies in wall-less plant cell protoplasts (Durand-Smet et al., 2014). Moreover, we find that the removal of the cytoskeletal filaments causes significant reductions in the dissipated energy upon indentation. The results also confirm that turgor pressure is the dominant resisting component to compression. Furthermore, AFM results and our analysis of the micro-indentation tests confirm that the high internal pressure in hypotonic conditions stresses the CW, effectively stiffening it, as predicted in (Cosgrove, 2016).

## 4. Materials and methods

### 4.1. Cell cultures

A cell culture of *Nicotiana tabacum* Bright Yellow-2 (BY-2) was provided by the Leibniz Institute (DSMZ, Braunschweig, Germany). The cell suspensions were transferred in fresh media every 2 weeks. 50–300 ml cell aliquots in 100 ml to 1 L flasks were kept on a rotary shaker at 130 rpm at 23–25 °C. The growth media comprised of a Linsmaier & Skoog (LS) medium with vitamins (HIMEDIA- PT040) and 3% (w/v) sucrose at a pH of 5.8, supplemented with 1 μM 2,4-dichlorophenoxyacetic acid (2,4-D), 1 μM a-naphtaleneacetic acid, and 1.46 mM $KH_2PO_4$. All chemicals were purchased from Sigma-Aldrich (St. Louis, MO). BY-2 marker lines expressing GFP-tubulin $\alpha$ that visualises MTs (GFP-BY2- $\alpha$), and GFP-AtFim1 to visualise AFs (GFP-BY2-F) were purchased (Riken BRC, Ibaraki, Japan) and cultured in modified Murashige and Skoog media (M0222, Goldbio, St. Louis, MO), supplemented with 0.2 mg/L 2,4-Dichlorophenoxy-acetic acid sodium salt monohydrate. These cultures were maintained on a rotary shaker at 130 rpm at 27°C and transferred weekly in 95 ml aliquots with 3 ml transfer volume in 300 ml flasks. The marker lines were only used for visualisation of the effects of drug treatments on the cytoskeletal filaments.

### 4.2. Cell treatments

To study the effects of depolymerising AFs or MTs in hypotonic conditions, cells were extracted from culture and introduced in growth media solutions containing either 250 μM LatB or 50 μM oryzalin for 2 min. To study the same treatments in hypertonic conditions, 250 μM LatB or 50 μM oryzalin diluted in 1M sorbitol were used to treat the freshly extracted from culture cells for 2 min. Control experiments were conducted in LS growth media, while for the pure plasmolysis experiments without any additional drug treatments, cells were extracted from culture and immersed in 1M sorbitol solutions. In both cases, after 2 min of exposure the cells were subjected to the selected experiments. The immediate effects of plasmolysis after exposure to sorbitol treatments were observed with light microscopy and with the optical microscopes embedded with the mechanical testing setups (Figures 1–3). The immediate AF or MT network disruption after the short treatment times were suggested through study of the GFP-BY2 marker cells using CLSM. Specifically, we observed unstained GFP-BY2 cells after 2 min of incubation time in each treatment (Figure 1) and concluded that this exposure time is enough to disrupt their cytoskeleton but not enough to cause other observable effects in the protoplasm. For the AFM tests and micro-indentation tests, cells were extracted from their culture in normal media, treated for 2 min in each solution, and subsequently placed on coated glass substrates to be tested mechanically. Both types of mechanical testing were conducted immediately after the cells were placed on the substrate.

### 4.3. Microscopy observations

To measure the CW thickness, super-resolution images of wild-type BY-2 cells stained with calcofluor white 0.005% (w/v) were acquired on an LSM980 CLSM (Zeiss, Oberkochen, Germany) through the Airyscan 2 SR system. Cells extracted from culture were immersed in staining solution for 2 min, and imaged immediately. A 63 × oil immersion objective (NA = 1.46) was used, and Z-stack images were acquired with 0.13 μm step-size and 0.034 × 0.034μm pixel-size. The Airyscan super resolution mode, coupled with a high NA objective allowed higher than diffraction-limited resolution in both *xy* and *z* planes. The measurement module of Imaris 9.7 (Bitplane) was then used to measure the CW thickness on multiple z planes per cell ($N = 61$) (Figure S7). Unstained GFP-BY2 marker cells in normal growth media, after 2 min of incubation in media-based 250 μM LatB or 50 μM oryzalin solutions, were imaged observed under a CLSM (SP5 II, Leica Microsystems, Wetzlar, Germany) using a 63 × water immersion objective (NA 1.2) to acquire Z-stacks. Light microscopy observations using an AxioScope A1 (Zeiss, Oberkochen, Germany) allowed length and width measurements of the cells. All image analysis was carried out in ImageJ (http://rsb.info.nih.gov/ij/).

### 4.4. Mechanical testing

We tested the mechanical properties of the cells in two different osmotic conditions, in 1M sorbitol and in growth media, and in three drug treatments in conjunction with each osmotic condition, no added drug, 250 μM LatB or 50 μM oryzalin. Cells were exposed to each of the test solutions at maximum for 15 minutes. The micro-indentation tests were performed using a FT-MTA02 system equipped with FT-S1000-LAT (liquid design) sensing probes with a 50 × 50 μm$^2$ square tip (FemtoTools AG, Zurich, Switzerland) and an optical microscope. Data from indentations were position-corrected to account for contributions of the system stiffness. Microscope glass slides (AmScope, Irvine, CA) were cleaned with isopropyl alcohol, surface activated with a high-frequency generator for 2 minutes (BD-20A, Electro-Technic Products,

Chicago, IL), and spin-coated with 0.5 ml poly-l-lysine (SUSS MicroTec, Garching, Germany). Cells extracted from culture and treatments were pipetted on the coated glass slides, washed several times with the selected treatment solution to effectively decluster them, and keep only those adhered to the substrate. One to three millilitres of the selected solution was added on top of the washed and diluted cells, and force-controlled indentations of up to 900 μN were conducted by immersing the sensing probe in liquid. The corresponding average indentation depth was $13.45 \pm 0.66$ μm (mean ± standard error). In the plots in Figures 3d and 5, each point corresponds to the indentation of an individual cell.

Short-range nano-indentations to evaluate the properties of the CW were conducted with AFM (Asylum Research, CypherES, Goleta, CA). For the AFM tests, we used custom tips with a silicon dioxide spherical particle (1 μm diameter) on a silicon cantilever with a stiffness of 16 N/m (Novascan, Boone, IA). Sample preparation was identical to that of the micro-indentation tests. We conducted force-controlled indentations to 15–70 nN and applied the Hertz model to calculate the indentation modulus, $E$, and a linear fit to calculate the initial stiffness of the cell wall, $k$. Each point in Figure 3d corresponds to an indentation test. We conducted multiple indentations for a given cell and tested a minimum of 5 cells.

### 4.5. Analysis

To subtract the sensor compliance, reference indentations on coated glass surfaces were conducted. The last 1 μm of indentation data is fit to a line, and the slope is taken to be the sensor stiffness, $S$. All micro-indentation experimental data on cells are then transformed by

$$\delta_{\mathrm{corrected}} = \delta - \frac{F}{S}, \qquad (16)$$

where $\delta_{\mathrm{corrected}}$ is the corrected indentation depth, $\delta$ is the measured indentation depth, and $F$ is the measured force. We determine the initial point of contact using a force thresholding method (Routier-Kierzkowska et al., 2012). The first 1 μm of filtered indentation data after the selected contact point are fit to a line, reported as the cell stiffness in that experiment, corresponding to one data point in Figure 3d. A Kolmogorov–Smirnov statistical test is used to compare the ECDFs of initial cell stiffness in each tested treatment. Sampling for the generative statistical modeling was performed using the Stan (Team, 2019) package within the Python programming language (https://www.python.org/). The optimisation to find the MAP parameters was performed using the SciPy (SciPy 1.0 Contributors et al., 2020) optimisation package, which contains a function to implement the trust-region interior point method described by Byrd et al. (Byrd et al., 1999). Visualisations were created using Altair (VanderPlas et al., 2018).

AFM nano-indentation data was processed using Asylum Research software (AR 16.10.211) in Igor Pro 6.3. The software was used to identify the contact point and extract a Young's modulus through the application of the Hertz contact model. The first 10% of the maximum force indentation data was fit to a line, and the average slopes ($k_{\mathrm{CW, GM}}$, $k_{\mathrm{CW, GM-MT}}$, $k_{\mathrm{CW, GM-AF}}$, $k_{\mathrm{CW, PS}}$, $k_{\mathrm{CW, PS-MT}}$ and $k_{\mathrm{CW, PS-AF}}$) in each test condition are used in the ratios $R$ in Equations (12)-(15).

### Acknowledgements

The authors thank Dr. Wai Pang Chan for obtaining confocal microscopy images, Dr. Takato Imaizumi, Dr. Josep Vilarrasa-Blasi and Dr. Luca Bonanomi for the helpful discussions.

**Financial support.** This work was supported by the University of Washington (E.R.); and the Resnick Sustainability Institute at Caltech (C.D.).

**Conflicts of interest.** No conflicts of interest declared.

**Authorship contributions.** E.R., L.G. and C.D. conceived and designed the study. E.R., L.G., R.M., Q.L. and G.S. conducted data gathering. E.R. and R.H. maintained the cell lines. L.G. performed data analyses and statistical analyses. E.R. and L.G. wrote the article and all authors edited it.

**Data availability statement.** The data reported in this study are available from the corresponding author upon request, and the code will be shared on https://github.com/lginsberg3/single-cell-biomechanics after publication.

**Supplementary Materials.** To view supplementary material for this article, please visit http://doi.org/10.1017/qpb.2021.15.

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
