## [Reviewer Report · Author comment: Cell wall and cytoskeletal contributions in single cell biomechanics of *Nicotiana tabacum*
— R0/PR1]

Dear Editors, 

Please find enclosed the manuscript titled “Cell wall and cytoskeletal contributions in single cell biomechanics of Nicotiana tabacum” By L. Ginsberg, R. McDonald, Q. Lin, R. Hendrickx, G. Spigolon, G. Ravichandran, C. Daraio and E. Roumeli submitted for publication as an Original research article in Quantitative Plant Biology. 

In this work, we investigate the contributions of the cell wall (CW), cytoskeleton and cytoplasm on the biomechanics of Nicotiana tabacum (bright yellow 2, BY2). We apply a multiscale biomechanical assay to single cells from a suspension culture. This assay includes nano-indentations through atomic force microscopy and micro-indentations through a micromechanical system. Based on our experimental results we propose a generative statistical model for the cell, which allows us to deconvolute the relative stiffness contributions from the CW, microtubules (MTs) and actin filaments (AFs) of the cytoskeleton and the (rest of the) cytoplasm. Our analysis provides evidence that the cytoskeleton significantly contributes to the mechanical response of BY2 cells in compression, while confirming that turgor pressure is the most significant contributor to the stiffness response of turgid cells under compression. 

Our work aims to contribute to improving our fundamental understanding of the mechanical properties of plants at a cellular and sub-cellular level, and specifically, to elucidate the mechanical role of components connecting to the cell wall.

Thank you for considering our contribution. We look forward to hearing from you.

Sincerely,

Eleftheria Roumeli

Assistant Professor of Materials Science and Engineering

University of Washington

---

## [Reviewer Report · Review: Cell wall and cytoskeletal contributions in single cell biomechanics of *Nicotiana tabacum*
— R0/PR2]

*Comments to Author*: The purpose of the work presented here was to assess the contribution of the cytoskeleton on the stiffness of plant cells, in particular tobacco BY2 cells.

The authors present a comparison of two techniques for measuring forces in BY2 cells: the micro-compression of cells using a flat indenter, vs nano-indentation with AFM. Such a comparison as never been conducted and is interesting in itself. The authors performed measurements both on turgid or plasmolyzed cells, which provides an excellent entry point for comparing the techniques.

Despite these promising ideas, the realisation of this work and the data interpretation present important shortcomings. Before publication, the authors would have to address the following major issues:

1) Some of the results showing a statics difference between groups of data are not fully convincing. In Fig. 3B, it is evident that there are much less data points collected for the growth medium conditions (GM, about 10 data points) than for the other conditions (GM-MT and GM-AF, each about 30 data points). Moreover, the values reported for the GM conditions range between 1 and 11 MPa, while the range for GM-MT is quite similar (about 1 to 16 MPa). If there was as many data points for GM as for GM-MT, it is possible that their distributions would be similar.

The same issue occurred for other measurements (fig 5B, low number of data points for GM-MT and for PS-MT).

More data points should be collected in the cases mentioned above. If it occurred that the difference between conditions was not significant, a large portion of the manuscript should be re-writen (results, conclusions, abstract).

2) Turgid cells (in GM) present a very large range of young’s moduli (2 to 11 MPa) and stiffness values (2 to 21 N/m) (fig3 and fig5), which is quite in contradiction with previous literature. However, the authors do comment on this. It is possible that some of the cells are not fully turgid, which can happen if the growth medium evaporated during the experiments, leading to an increase in the GM osmotic potential and a decrease in the cell turgor pressure. For this reason, (Weber et al. 2015) monitored the osmolarity of the growth medium before/after experiments.

3) The authors report the cells to be stiffer when the microtubules are depolymerized (fig 5B). However, this counterintuitive result is not addressed.

4) There is only one short mention of (Weber et al 2015), which is very surprising, given how relevant Weber's work is to the present manuscript. Using FEM modeling, CFM and osmotic experiments, they showed the force measured with CFM in BY2 cells was proportional to the turgor pressure and to the cell radius. The authors should refer to Weber's paper (e.g. in the intro, circa line 66) and compare their experimental results and mechanical model in details (e.g. in the results, circa line 172).

5) The authors observe, as in other studies (e.g. Weber et al. 2015, Routier-Kierzkowska et al. 2012) a higher apprent stiffness of turgid cells compared to plasmolyzed ones. The only interpretation given by the authors is that the cell wall changes its mechanical properties under tension (e.g. line 280-281). However, a simpler explanation was already provided by the FEM models of Weber et al. 2012 and Routier-Kierzkowska et al. 2012, which reproduced this effect without introducing strain-stiffening properties to the cell wall. Generally speaking, a membrane or a shell under tension will appear stiffer, but that doesn't mean it is stiffening in response to stress. To make an analogy: the skin of a drum is harder to deform (i.e. it appears stiffer) as it is tightened (as the tension is increased), even if the skin itself keeps the same elastic modulus. The tension in a cell wall is caused by turgor pressure, but the principle is the same as in the case of the drum skin.

6) Similarly, AFM results can be interpreted without strain stiffening of the cell wall. The indentation depth of the AFM tip was about 20% of the depth of the cell wall (Line 128-130). Surely, this should be sufficient to bend the wall when indenting plasmolysed cells, as opposed to compressing the cell wall locally when indenting turgid cells. This could explain why plasmolyzed cells have a much lower young’s modulus than turgid ones.

Minor points:

1) Figure 2: I find the schematic representation confusing. In particular, why are MTs represented as more parallel to each other than AFs? This is not the case in the observations.

2) Line 118-119: I don't understand what the authors mean here and the supplementary material were not provided. Surely, the osmotic pressure of the growth medium (GM) should be lower than the one of the plasmolizing solution (PS)?

3) Line 169: “The evident increase in stiffness observed in cells tested in a solution of higher osmotic pressure”. It is the opposite: higher osmotic pressure of the solution = lower hydrostatic pressure of the cell = softer cell. For example, pure water has a null osmotic pressure, which induces a maximal hydrostatic pressure within the cell.

4) Fig. 5A: why use a different representation of the same data as in B? It is quite confusing and unecessary.

5) Fig. 5B: Why not show here the significance of the statics difference, as in fig. 3?

---

## [Reviewer Report · Review: Cell wall and cytoskeletal contributions in single cell biomechanics of *Nicotiana tabacum*
— R0/PR3]

*Comments to Author*: This paper addresses the role of turgor pressure, cytoskeletal elements and the cell wall itself in determining the mechanical properties of the plant cell wall. These interesting questions are highly relevant to those working on plant biomechanics and development given the mechanical nature of cell structure and expansion. The authors use a combination of indentation methods, pharmacological and osmotic treatments, and mathematical modelling to tease out the individual contributions to cell wall mechanics of different cellular components. As many measurement techniques give variable results across experiments, the authors attempt to bring together measurements across different scales providing a substantial contribution to the field.

The paper is generally very clearly written but there are a few areas in which I think the manuscript could be improved.

1. Cell wall thickness measurements are carried out here using Calcafluor staining and super-resolution confocal images and is relevant to understanding the extent to which indendations are likely to be influenced by turgor pressure. The method used is not particularly standard compared to other publications, however most methods of measuring cell wall thickness have issues. The measurements in the present manuscript (0.79um) are considerably higher compared to TEM measurements of BY-2 cells (0.1um) (Sabba et al., 1999 Int. J. Plant Sci. 160:275). It may be that fixatives during TEM sample preparation dehydrate the wall and give rise to this discrepancy, however, some studies have found increased wall thickness in TEM measurements compared to freeze-fractured SEM images that should preserve hydration. (see Derbyshire et al. 2007, J. Exp. Bot. 58:2079; Haas et al. 2020, Science, 367:1003). To help evaluate the validity of these measurements, it would be helpful to provide some example confocal images in which cell wall thickness was measured, and to provide some additional discussion of the potential limitations/variability in measurements of cell wall thickness.

2. The concentration of latrunculin B used (250uM) seems quite high (though admittedly treatment times are very short). Was there a rationale for this high concentration? In other publications with BY-2 cells, ranges of 0.1-10uM appear typical. There could also be a bit more explanation in the introduction of the effects that oryzalin and lat B have on the cytoskeleton and in general on plant cell/tissue function.

3. In AFM experiments, lat B treatment appears to substantially reduce stiffness in plasmolysed cells (fig 3). This seems surprising and there could be a little more discussion about the significance or interpretation of that result.

4. I am not qualified to assess the details of the statistical model that was constructed so will leave that to other reviewers but I found the explanation generally quite clear. I would suggest though that for the general reader additional explanation is given to help understand why two different models are constructed and the significance of the opposite results for the role of MTs versus actin in each model.

Minor points

5. Why are deeper indentations carried out on the plasmolysed cells – are they more reliable?

6. Sentence starting ‘The stiffness contributions’ in line 216 took me a while to understand. Perhaps it could be made a little clearer?

7. Mistake in line 220: kCW,hyper repeated twice

8. Line 252, ‘is’ should be ‘are’

9. Line 297, missing brackets

---

## [Reviewer Report · Recommendation: Cell wall and cytoskeletal contributions in single cell biomechanics of *Nicotiana tabacum*
— R0/PR4]

*Comments to Author*: Both the Reviewers found the problem addressed by this manuscript interesting and significant for biomechanics of plant cell. However, they also both found numerous issues that need to be addressed by the Authors before the manuscript can be accepted for publication. Please note that the Reviewers, having a different scientific background, point to different issues and therefore addressing all of them should improve this manuscript.